# Introduction of Native Submerged Macrophytes to Restore Biodiversity in Streams

**DOI:** 10.3390/plants13071014

**Published:** 2024-04-02

**Authors:** Lucas Van der Cruysse, Andrée De Cock, Koen Lock, Pieter Boets, Peter L. M. Goethals

**Affiliations:** 1Department of Animal Sciences and Aquatic Ecology, Faculty of Bioscience Engineering, Ghent University, Coupure Links 653, 9000 Ghent, Belgium; andree.decock@ugent.be (A.D.C.); koen.lock@ugent.be (K.L.); pieter.boets@oost-vlaanderen.be (P.B.); peter.goethals@ugent.be (P.L.M.G.); 2Provincial Centre of Environmental Research, Godshuizenlaan 95, 9000 Ghent, Belgium

**Keywords:** submerged aquatic plants, ecological restoration, water quality, river biodiversity, lotic ecosystems, nature-based solutions, nature restoration

## Abstract

Streams are biodiversity hotspots that provide numerous ecosystem services. Safeguarding this biodiversity is crucial to uphold sustainable ecosystem functioning and to ensure the continuation of these ecosystem services in the future. However, in recent decades, streams have witnessed a disproportionate decline in biodiversity compared to other ecosystems, and are currently considered among the most threatened ecosystems worldwide. This is the result of the combined effect of a multitude of stressors. For freshwater systems in general, these have been classified into five main pressures: water pollution, overexploitation, habitat degradation and destruction, alien invasive species, and hydromorphological pressures. On top of these direct stressors, the effects of global processes like environmental and climate change must be considered. The intricate and interconnected nature of various stressors affecting streams has made it challenging to formulate effective policies and management strategies. As a result, restoration efforts have not always been successful in creating a large-scale shift towards a better ecological status. In order to achieve an improved status in these systems, situation-specific management strategies tailored to specific stressor combinations may be needed. In this paper, we examine the potential of introducing native submerged macrophyte species to advance the restoration of stream ecosystems. Through successful introductions, we anticipate positive ecological outcomes, including enhanced water quality and increased biodiversity. This research is significant, as the potential success in restoring stream biodiversity not only represents progress in ecological understanding but also offers valuable insights for future restoration and management strategies for these vital ecosystems.

## 1. Introduction

Streams, like other aquatic ecosystems, support a disproportionately high amount of biodiversity when compared to their global surface area [1,2]. This biodiversity consists of a wide variety of taxonomic groups, including but not limited to: microorganisms, algae, macrophytes, insects, nematodes, mollusks, crustaceans, fish, amphibians, and mammals [3,4]. Therefore, these systems can be considered crucial biodiversity hotspots. The inherent importance of this biodiversity can be illustrated by examining the ecosystem services provided by these systems, since sustainable ecosystem functioning is dependent on the diversity in functional traits between species [5]. Stream ecosystems provide essential resources, including food, drinking water, and energy production, while also performing key ecological processes such as climate regulation, nutrient cycling and water purification [1,6,7]. In addition to these tangible benefits, stream systems can offer non-material services as well, such as recreational and educational values [1,6,7]. As providers of such an extensive list of ecosystem services, stream systems are irreplaceable from both ecological and economic viewpoints [8].

Despite this importance, stream ecosystems are considered to be among the most threatened ecosystems on the planet [9]. This threatened status results from a range of stressors, predominantly stemming from human activities and their impact on these ecosystems [2,10]. In addition, the inherent property of stream systems to be directly connected to their catchments causes them to be even more vulnerable to land use influxes such as agricultural runoff [9]. Consequently, freshwater ecosystems, and especially stream systems, have deteriorated extensively on a global scale [11]. As a result, freshwater systems have witnessed a disproportionate decline in biodiversity compared to other ecosystems [6,12]. This trend is problematic. Research by Vörösmarty et al. in 2010 found the level of impact on stream biodiversity to be highly correlated with levels of threat to human water security when compared spatially [13], once again demonstrating that the ability of freshwater ecosystems to provide ecosystem services is inherently linked to their biodiversity [5]. Therefore, it is crucial to protect and restore stream ecosystems under pressure to a good ecological status and preserve this status to ensure sustainable ecosystem functioning.

Thus far, stream restoration efforts have not consistently succeeded in sustainably improving ecosystem health and functioning [14], often focusing on physical stream characteristics regardless of which stressor combination is causing degradation [15]. Some of the challenges that must be accounted for in the future include the complexity of site-specific conditions, difficulty in considering all relevant abiotic and biotic variables, and obstacles to incorporating appropriate spatial and temporal scales for restoration [14]. To overcome these hurdles, there is a pressing need for more integrated restoration strategies to develop a more holistic understanding of river ecosystem interactions [14,16].

Even when restoration efforts succeed in improving water quality and/or hydromorphological conditions, the recovery of the associated flora and fauna is not ensured [14,17,18]. This recovery also relies on the proximity and connectivity of viable source populations from which species may recolonize these newly restored sites [17]. The reestablishment of native macrophyte species, for example, may be halted due to dispersal limitations or lack of diaspore reservoirs (or both), despite apparently suitable habitats being available [19]. In situations where natural recolonization is lacking, reintroduction efforts thus emerge as a potential solution [20]. Sustainable rehabilitation of stream ecosystems through reintroduction of macrophytes can be achieved through a meticulous selection of species for introduction, with an active acknowledgment and consideration of the prevailing on-site conditions in the reintroduction sites [21]. While some successful examples exist [21,22,23], these types of (re)introductions for macrophytes are scarce and have only been successfully performed in a handful of studies. Hence, significant knowledge gaps persist, which hamper establishment of a framework for the broader implementation of such practices.

In this paper, we advocate a novel, nature-based approach to restore shallow stream ecosystems. By introducing native submerged macrophytes in streams that meet the minimal chemical water quality requirements for macrophytes, this approach can instigate the revitalization of these shallow stream systems. Anticipated ecological benefits encompass improvement in water quality and the facilitation of increased biodiversity (e.g., macroinvertebrates) in these ecosystems [22]. By identifying key drivers determining establishment success for submerged macrophyte species beforehand, well-substantiated decisions regarding both site and species selection can be made. This knowledge can be acquired through a broad approach, integrating extensive analyses on preexisting monitoring and species distribution data, additional field studies and ex situ experimental setups.

In the following segments, we will detail the current threats to freshwater ecosystems, after which we will discuss policies, water management, and restoration efforts. Finally, we will highlight the role introductions of submerged macrophytes may hold in future freshwater restoration. In addition, we give a broad overview of the way in which stream restoration efforts may capitalize on these benefits in the future.

## 2. Threats to Freshwater Ecosystems

To establish a foundational understanding of the challenges faced in the restoration and preservation of stream ecosystems, it is essential to explore which general threats are imposed on freshwater ecosystems. The largest global threats to freshwater systems can be classified into six main pressures: water pollution, overexploitation, habitat degradation and destruction, alien invasive species, hydromorphological modification, and finally climate change, which is superimposed on the first five pressures [2,9,10].

**Water pollution** remains a pervasive problem. Although some industrialized countries have made progress in reducing industrial and civil pollution, growing threats from pesticides and excessive nutrient enrichment through land use runoff remain [10]. Especially stream systems are prone to these influxes through their direct connection with their catchments. Therefore, the impact of land use runoff on stream systems must be assessed on a large spatial scale. Anthropogenic activities on the landscape scale impact stream habitats, water quality, and communities through various pathways [24]. For example, the nutrient status in European rivers has been identified as one of the main drivers shaping aquatic plant communities [25]. Additionally, the concern over freshwater pollution now includes the emerging issue of microplastic contamination as well [2].

**Overexploitation** of freshwater systems is a major driver of biodiversity loss, mainly affecting fish, reptiles and some amphibians. By excessively removing certain species from their habitats for consumption or through bycatch, population sizes can become significantly diminished, which can eventually cause species extinctions [2,10].

**Habitat degradation** can be the result of various anthropogenic practices, including direct impacts like excavation and indirect impacts from changes within the drainage basin, such as deforestation [10]. Degradation caused by human-induced disturbances can often be directly linked to lower species richness [26,27]. By altering the physical structure and quality of freshwater habitats, the species that depend on them are directly impacted.

**Invasions by alien species** are considered a global threat to freshwater biodiversity [28]. This threat is exacerbated by increasing trade, unintentional species transport, and the construction of canals connecting biogeographic regions [29]. Alien species, with the ability to invade especially ecosystems where habitat degradation places native populations under stress, can exert large-scale effects that further disrupt these systems [30,31]. This disruption may result from superior competitive capabilities compared to native species or the introduction of predation on species not previously exposed to such threats [32,33]. Furthermore, many alien plant species in freshwater systems form monotypes, diminishing local biodiversity and drastically altering habitat structure, nutrient cycles and food webs [34].

**Hydromorphological modifications** are most intrusive in stream systems. Due to dam construction, the impacts of expanding hydropower, and river rectifications, river flow is altered globally [2]. By altering the hydromorphology of stream systems in such a drastic way, habitats and the species depending on them are impacted gravely. For macroinvertebrates, for example, the number of taxa that occur within a community is highly dependent on stream velocity, and exceeding certain tipping points can be devastating to local macroinvertebrate diversity [35] and traits [36].

Finally, **climate change** poses significant threats to ecosystem functioning and biodiversity by altering temperatures, elevating atmospheric and aquatic CO_2_ levels, and increasing the frequency of extreme weather [37]. Freshwater systems in particular face heightened vulnerability to climate change due to its influence on global temperatures, which affects both the temperature in these systems directly (qualitative aspects) and the water availability in these systems (quantitative aspects). Through the alteration of precipitation and evapotranspiration patterns, the severity of droughts and peak floods will likely increase [38]. As a consequence, the distribution and functionality of freshwater systems is being affected [39,40]. Effective management strategies adapted to the changing climate are crucial for preserving and restoring biodiversity in the face of these environmental challenges. Furthermore, the fragmented and often isolated nature of freshwater systems further intensifies their susceptibility to rapid environmental change. The ability of freshwater species to disperse may be insufficient for relocation as the climate changes. Additionally, recolonizing habitats lost due to extreme weather events may prove difficult [41].

Increased nitrogen deposition, global warming, and changing precipitation and runoff patterns are all threats superimposed by climate change on the freshwater-specific threats already discussed [2,10]. As a result, all of these threats should be considered both individually and as stressors with varying interaction effects. For example, a meta-analysis by Birk et al. (2020) investigated the responses of freshwater ecosystems on both climate and land use stressors and found varying outcomes in stream systems depending on the specific stressor combinations present and the biological response variables used [42]. These findings underscore the necessity for situation-specific management strategies, particularly in stream ecosystems where interactions between stressors exhibit a nuanced and context-dependent nature.

## 3. Policies, Water Management, and Restoration Efforts

In response to the escalating threats facing freshwater ecosystems globally, numerous influential policies have been established since the late 20th century. The European Union took a pioneering step by introducing the Water Framework Directive 2000/60/EC (WFD) in 2000, aiming to rejuvenate and safeguard aquatic habitats while enhancing water quality [43]. Beyond Europe, various nations have implemented pivotal measures such as the United States Clean Water Act (CWA), Australia’s Water Quality Guidelines, and China’s Water Pollution Prevention and Control Law [44,45,46]. Additionally, international frameworks like the UN Sustainable Development Goals 6 and 15 (SDG 6 and SDG 15) and initiatives supported by the Global Environment Facility (GEF) contribute to the global commitment to freshwater restoration on broader scales [47,48]. In this collective effort, policies like the Nature Restoration Law (NRL) in Europe and the Blue Deal in Flanders play a crucial role, serving as examples of policies on national and regional scales [49,50]. These examples highlight the diverse strategies adopted worldwide to address the multifaceted challenges of freshwater ecosystem health. Although these efforts clearly have had positive effects on biodiversity, in recent years, they have appeared to offer diminishing returns [18]. Approximately 60% of European surface water bodies still lack good or better ecological status, with rivers and transitional waters scoring worse than lakes and coastal waters [12].

The reason these efforts have not consistently succeeded in elevating stream systems to a higher ecological status is likely multifaceted. Historically, the focus of stream conservation and restoration efforts was mainly on ensuring an ample supply of drinking water, and thus only structural and physical properties of these systems were considered [51]. Furthermore, restoring and preserving stream ecosystems is especially difficult due to their direct connection to their catchments. Considering influences on a regional scale may be necessary to ensure successful restoration [51]. Unfortunately however, approximately 70% of river segments worldwide (measured by length) lack protected zones within their upstream catchment areas [52]. Implementing restoration measures on a larger scale is not an easy task, as the implementation of effective measures on any scale is often further complicated by competition among stakeholders with differing interests [10].

Acknowledging these limitations, the European Green Deal and the EU Biodiversity Strategy for 2030 have become pivotal in shaping a more effective approach. A notable component is the Nature Restoration Law (NRL), scheduled for a final vote in early 2024. The NRL mandates EU member states to implement restoration measures on at least 20% of land and marine areas by 2030, extending to all ecosystems requiring restoration by 2050 [53]. This includes specific targets for the restoration of streams and rivers.

As a result of these various policies, significant efforts have been dedicated to counteracting the degradation of stream ecosystems. Feld et al. reviewed the most common restoration practices [14]: the implementation of riparian buffer zones, the introduction of in-stream habitat structures, and the removal of weirs and dams.

Riparian buffer zones are designated sections of land that encapsulate streams of interest. These zones are designed to mitigate the impacts of intensive agricultural land use by retaining plant nutrients, fine sediments, and toxic substances from entering streams from adjacent areas [14]. These retaining qualities are in part attributed to riparian vegetation, which can be grassland, trees or mixed vegetation [54]. The effectiveness of these buffers is contingent upon their width and length, with a range of 15–30 m in width and over 1 km in length proving most effective [14,55]. While riparian buffer zones have demonstrated positive influences on abiotic variables, such as habitat diversity and nutrient retention, the evidence for significant biological recovery remains scarce [14].

Creating additional in-stream habitat structures, including the introduction of large woody debris, boulders, artificial riffles, and gravel, aims to promote biological diversity through the enhancement of physical diversity. The increased habitat heterogeneity resulting from such measures is believed to increase the potential for biodiversity in ecological theory [15]. However, despite most studies achieving higher physical habitat heterogeneity, significant increases in biodiversity are rare [14,15]. These findings challenge the assumption that physical heterogeneity alone is a sufficient driver of stream biodiversity.

The removal of weirs and dams targets the restoration of longitudinal connectivity, aiming to benefit instream fauna, such as migrating fish. This approach has demonstrated positive effects on hydromorphological conditions and sediment particle size, as well as enhancing effects on fish habitats [14,56]. However, short-term adverse effects, such as the mobilization of fine sediments, can initially eclipse any positive effects that usually pay off over longer time scales [57]. For weir and dam removal in particular, spatial and temporal considerations are crucial. Removing weirs close to the spawning season, for example, may limit downstream availability of spawning habitats that season [14].

While riparian buffer zones, in-stream structures, and the removal of weirs and dams are commonly employed in stream restoration efforts, the limitations of these measures highlight the complexity of achieving long-term success. Determining whether successful restoration efforts result in sustainable ecological improvement is difficult as well, because the duration of most studies on stream restoration efforts is too limited to provide a firm understanding of long-term ecological implications [14]. Nevertheless, the challenges encountered in past stream restoration efforts can be evaluated to highlight certain areas that can be strengthened for future endeavors. For instance, a more comprehensive approach is needed to better understand the influence of catchment land use on hydrological and geomorphological processes. Additionally, the consideration of unique site-specific conditions and ensuring a thorough evaluation of relevant abiotic and biotic variables can contribute to more effective restoration strategies. Lastly, the incorporation of appropriate spatial and temporal scales is another practice that can be refined based on lessons from previous experiences. Recognizing these opportunities for improvement underscores the importance of adopting more integrated restoration strategies to foster a deeper understanding of interactions within river ecosystems [14,15].

Due to these limitations, approaches to freshwater ecosystem restoration that mainly focus on water quality and habitat restoration have not always been able to fully restore biotic communities. Nevertheless, it is crucial to stress the importance of these restoration efforts. By improving water quality, reverting hydromorphological alterations to natural flow, and minimizing land use influxes; these management practices contribute directly to the restoration of favorable abiotic conditions for native species. However, they do not directly contribute to the rehabilitation of native populations of flora and fauna themselves. The flawed assumption that habitat restoration alone may guarantee the return of native organisms reveals a significant disparity between anticipated and actual biological outcomes. Therefore, reintroduction of native species may be a useful additional management solution that is able to take full advantage of the improvements achieved through more conventional approaches. It is important to stress that reintroduction success is dependent on these previous improvements, because if the original stressors that caused species to disappear remain unaltered, sustainable reintroductions are unlikely to succeed [16,58].

In the following segments, we will discuss how the reintroduction of native submerged macrophyte species may be utilized to further progress the restoration of these freshwater ecosystems.

## 4. Macrophyte Introduction: Macrophytes as Primary Ecosystem Engineers

Given the complexity of the threats documented by Dudgeon et al. (2006) and Reid et al. (2019) [2,10], as well as the varying biological response to situation-specific interaction effects [42], a one-size-fits-all approach to freshwater management through improving water quality and restoring habitats may be unrealistic. In this section, we discuss the potential of submerged macrophyte introductions, given minimal requirements of chemical water quality and in the absence of previous stressors, as an additional management solution for the restoration of stream ecosystems.

Macrophytes are crucial components of stream ecosystems, and their significance in these systems extends far beyond their physical presence. As primary producers, macrophytes contribute to nutrient cycling and water purification processes, and their community composition is influenced by species-specific responses to changes in water quality. As a result, they are very useful bioindicators [59,60] and key contributors to the assessment and monitoring of freshwater ecosystems. Their ability to accumulate heavy metals in their tissues makes them valuable indicators of contamination as well [61]. The recognition of macrophytes as biological quality elements by legislation (i.e., WFD, CWA) has further intensified scientific interest in this taxonomic group, leading to a surge in newly created indices and predictive models in recent decades [62,63]. Furthermore, macrophytes provide a variety of habitats for macroinvertebrates, zooplankton, and fish living in these ecosystems [64]. Without them, populations of macroinvertebrates and zooplankton are likely to be flushed out in stream ecosystems [65].

In rivers and streams, macrophytes actively shape these stream ecosystems by altering stream velocity and sedimentation dynamics, creating heterogeneity in hydromorphology. As a result, they are sometimes referred to as “primary ecosystem engineers of fluvial systems” [66]. Moreover, macrophytes contribute to increased water retention by increasing hydraulic resistance. These effects are limited to late summer months, and will never cause flooding to the extent it already naturally occurs in winter months [67]. Because of these combined benefits, the reintroduction of macrophytes can be a very effective management strategy to improve the ecological status of degraded stream systems [68].

Macrophyte introduction may in some cases not only be very effective, but necessary to ensure reestablishment. This is due to the fact that species sorting, the determination of species distributions and abundance through abiotic conditions, cannot always fully explain macrophyte occurrences. Recent analyses by Terrijn et al. (2021) emphasize that both dispersal limitation and species sorting through abiotic pressures contribute to macrophyte community structure variation in freshwater ecosystems in Flanders (Belgium) [69]. These results suggest that both favorable abiotic conditions, as well as the ability of species to reach these favorable habitats, are crucial to the rehabilitation of native populations in this region. It is the latter of these constraints that may be addressed by the proposed management strategy of introduction.

In the absence of upstream plant populations, colonization depends on species dispersal from other freshwater systems, often mediated by animals or humans. This process is inherently slow, with the probability of a species dispersing to a specific stream positively correlating with its regional abundance [21]. Consequently, even if physical and chemical habitat conditions are restored in streams, the natural colonization remains dependent on species population numbers in the region. Therefore, promoting recolonization through reintroduction can in certain cases be highly beneficial. Riis et al. demonstrated that reintroduction of macrophytes to streambeds is a sustainable means of stream rehabilitation, especially when species selection aligns with the physical conditions in the streams [21]. As a result, the introduction of submerged macrophytes may prove very effective in the effort to overcome dispersal limitations.

However, it is crucial to recognize that introductions are complex interventions influenced by various factors and that caution is warranted when performing them. If natural recolonization can occur solely through restoration that addresses the local drivers for species sorting, this is usually a more cost-effective approach. If not, however, it is crucial to make well-informed, data-driven decisions to optimize the likelihood of sustainable introduction outcomes. In the following section, we will delve into the lessons learned from past introductions, providing valuable insights into the considerations essential for success.

## 5. Considerations for Successful Macrophyte Introduction

The literature on macrophyte introduction efforts in stream ecosystems is scarce, and as a result, good practices are not yet well defined. This contrasts with introduction efforts for fish species, for example, which are much more common [70]. However, a handful of studies have already provided some very useful insights for macrophyte introduction [19,20,21]. We will discuss these recommendations in this segment.

The first and likely the most important step to sustainable introduction is the selection of suitable introduction sites. Riis et al. defines three essential rudimentary physical requirements for sustainable macrophyte growth and thus reestablishment in stream systems: shallow water (<1 m), moderate water velocity (<0.4 m/s) during plant establishment, and unshaded conditions [21]. These are good initial guidelines to already exclude a variety of stream ecosystems. However, more intricate knowledge of local conditions is usually needed to select appropriate sites. The introduction of submerged macrophytes into a site can only succeed when environmental conditions are favorable, necessitating a comprehensive understanding and knowledge of local abiotic factors during site selection [19]. In cases where native species are reintroduced after disappearing due to habitat degradation, it is important to note that these efforts can only be successful if the original causes of population decline have been eliminated [71]. Factors like hydromorphology, sediment quality, historical occurrences of native submerged macrophyte species and nutrient concentrations can all be crucial in site selection as well [21]. The significance of these factors depends on the specific local conditions and the species chosen for introduction. Also, accounting for the vulnerability of specific sites to climate change can be essential in ensuring the sustainability of introduced populations [20]. It is, for instance, sensible to prioritize the selection of introduction sites characterized by inherent resilience to peak floods and drought, given the anticipated increased frequency of these events due to climate change. Such resilience is crucial, as these phenomena can significantly impact the establishment and persistence of submerged aquatic vegetation [72].

Secondly, selecting appropriate species for reintroduction is crucial as well. The process can initially be guided by prioritizing species that are or were naturally present in the region [21]. Next, species-specific preferences and vulnerabilities must be considered in relation to current conditions in introduction sites. For example, *Potamogeton* spp. and *Myriophyllum* spp. are often selected because of their efficient light utilization through the accumulation of much of their biomass near the surface, as well as their ability to thrive despite periods of elevated turbidity and eutrophic conditions because of specialized root structures supporting growth [19,20,21]. Additionally, the capacity of species to secrete allelopathic compounds may too play a role in species selection, as it is an effective means to inhibit the growth of microalgae [20]. Finally, if possible, plants used for introduction should be obtained or cultivated from populations in close proximity to the introduction site. By applying this rule, specimens will be as genetically similar to the original population as possible [19].

Finally, in order to introduce macrophytes, there are three available methods: sowing, direct transplantation, and sampling followed by cultivation and introduction. Because sowing is often complicated by the scarce availability of seeds [20] and direct transplantation can result in a higher ecological cost in sampling sites, because more specimens need to be removed, we advocate for the latter strategy. Successful introduction requires careful consideration of the introduction site’s depth and flow rate to minimize the risk of biomass loss due to drag. Timing is crucial, and introductions must only be performed in periods where high flow rates or peak floods are unlikely [19]. As a result, it is ideal to introduce macrophytes early in the favorable season, as flow rates remain moderate and the onset of microalgae and filamentous algae has not yet occurred, which elevates potential competition effects [20].

Based on some of the considerations mentioned in previous segments, as well as some supplementary literature [49,50,73,74], a visual representation of a SWOT analysis has been added below (Figure 1) to summarize the main strengths, weaknesses, opportunities, and threats to this restoration approach.

## 6. Towards a Successful Implementation Strategy for Submerged Macrophyte Introduction

The current paper advocates for the introduction of macrophytes, highlighting their inherent benefits in nutrient cycling, water purification, and habitat provision. The deliberate choice of submerged macrophyte species is driven by strategic consideration. The introduction of submerged species specifically overcomes challenges associated with water management costs and mowing of other types of macrophytes. By opting for submerged species, we ensure a balanced approach that capitalizes on the inherent positive effects of macrophytes while mitigating specific management hurdles and conflicting interests with local stakeholders. Furthermore, although the literature on macrophyte introductions is scarce and often focuses on lakes, most available sources focus on submerged species [19,20]. The choice for these taxa is thus further substantiated by a desire to make well-informed considerations regarding introduction efforts whenever possible.

First of all, it is crucial to identify the key variables influencing species sorting in submerged macrophyte communities. To identify these variables in the streams selected for introduction, research should focus on likely drivers first, such as physicochemical variables, hydromorphology, river sediment quality, surrounding land use, and pesticide presence. This can be accomplished through a varied approach, integrating extensive analyses on preexisting monitoring and species distribution data, additional field studies and/or ex situ experimental setups.

Secondly, small-scale, well-monitored pilot introduction experiments should be performed with native submerged macrophytes, evaluating the effectiveness of such procedures for permanent recolonization. These introduction efforts, if successful, will overcome existing dispersal barriers and provide valuable insights into the positive ecological outcomes these types of introductions may have in the future. The insights gained from each of these introduction experiments can serve as a foundation for improving and optimizing subsequent introduction experiments in the following growth seasons. This iterative learning process can be crucial to refine methodologies and enhance the overall efficacy of future introduction initiatives.

In addition to assessing restoration success and individual plant growth, it is essential also to evaluate effects on chemical and ecological water quality, as well as hydromorphological variables. Furthermore, broader ecological effects, including the impact on macroinvertebrates, must be considered as well to estimate the effects on biota. Consistently monitoring this large number of variables over longer time periods is essential to estimate long-term effects and ensure the sustainability of these introductions. This continuous monitoring will provide critical data on the adaptability and resilience of the introduced species in their respective environments.

In order to ascertain the true impacts of introduction efforts, it is crucial to adopt a well-organized and rigorous methodology. A recommended approach is the before–after control–impact (BACI) methodology [75]. By comparing the introduction sites with equivalent non-introduction sites, likely located upstream, changes over time can be analyzed and measured more effectively. It allows for a thorough examination of the reintroduction efforts’ effects by distinguishing them from external factors. The BACI methodology provides a robust framework for assessing the true impact of introductions, contributing to the reliability of the findings.

Figure 2 provides an overview of how a research implementation strategy can be structured in order to optimize the success and sustainability of native submerged macrophyte reintroductions.

## 7. Conclusions

The persistent challenges faced by stream ecosystems demand innovative and adaptive solutions building on the progress of traditional restoration methods. By focusing on submerged macrophytes, the research anticipates positive ecological outcomes, including improved water quality and increased biodiversity. Potential implications and benefits of this approach offer insights that may be useful in addressing the broader goals of the Water Framework Directive. Also, further research and implementation of this approach may provide a framework for the development of similar management strategies in the future. Ongoing research in this field remains crucial in order to develop novel strategies to overcome the evolving challenges threatening stream ecosystems and their biodiversity.

## Figures and Tables

**Figure 1 plants-13-01014-f001:**
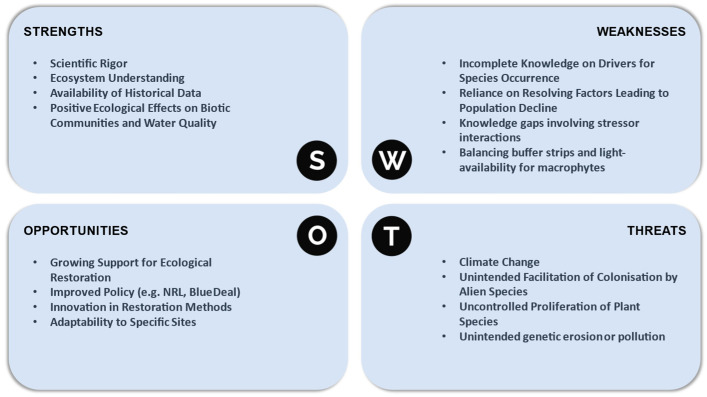
SWOT analysis based on the considerations for successful macrophyte introduction.

**Figure 2 plants-13-01014-f002:**
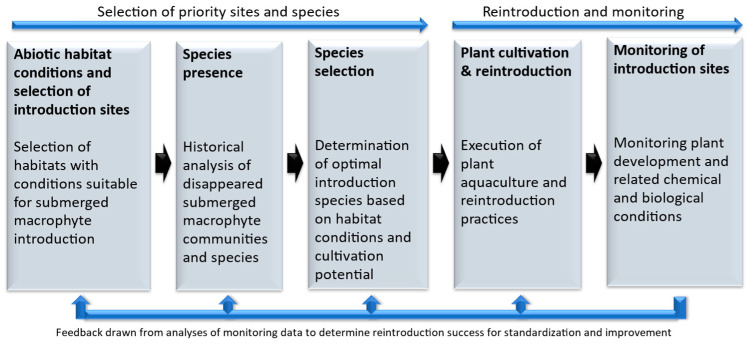
Broad overview of a possible research implementation strategy and structure.

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
