# Peer review of "Introduction of Native Submerged Macrophytes to Restore Biodiversity in Streams"

_plants, 2024, doi:10.3390/plants13071014_

Round 1

Reviewer 1 Report

Comments and Suggestions for Authors

Manuscript number: plants-2890529

Title: Introduction of native submerged macrophytes to restore biodiversity in streams

The manuscript was submitted to section Plant Ecology, special issue Macrophytes in Inland Waters: From Knowledge to Management Ⅱ.

I wonder why the Authors chose the article type as perspective paper. In my opinion, it is rather a review article that does not exhaust the issues contained in the title.

Section 1. Introduction

This section clearly lacks a review of the literature on stream biodiversity and its role in maintain ecosystem balance

Section 4. Macrophyte Introduction: Macrophytes as Primary Ecosystem Engineers

The literature review on the role of macrophytes in river ecosystems is very poor. The role of macrophytes in running waters is very well documented, especially their bioindication potential, which is reflected in the macrophyte water quality index implemented under the Water Framework Directive for monitoring the ecological status of streams and rivers.

5. Considerations for Successful Macrophyte Introduction

The Authors refer to macrophytes species, Potamogeton spp. and Myriophyllum spp. that are successfully used for restoration treatments, but here again the literature review is very poor. Specific examples (case studies) should be cited along with their results.

Based on the above, whether the SWOT analysis was prepared solely based on the literature cited in this section?

Considering the above,  I believe that the manuscript in its current form is not suitable for publication.

Reviewer 2 Report

Comments and Suggestions for Authors

Restore biodiversity in streams with native submerged macrophytes is a hot topic of substantial interest. Although this manuscript review some useful perspectives of biodiversity restoration in streams with native submerged macrophytes, I have some substantial concerns about the manuscript, including aspects of the presentation and some of the interpretations. Below I describe my major concerns followed by a list of more specific points.  

Major concerns:

The Abstract does not adequately indicate the general relevance of this work.  Although the first sentence indicated why this topic is important, and the last sentence should indicate the major implications for advancing ecological understanding.

The Introduction is fine at this status, but it is better if the authors can focus more about the introduction of native submerged macrophytes.

2. Threats to Freshwater Ecosystems: The largest global threats to freshwater systems can be classified into five main pressures is not good, I think Climate change should be integrated into the sixth main pressures.

3. Policies, Water Management and Restoration Efforts: Provide more Policies, Water Management and Restoration Efforts in stream restoration,especially of native submerged species. Furthermore, more regiones rather than Europe should be payed attention to.

4. Macrophyte Introduction: Macrophytes as Primary Ecosystem Engineers:

Add some information about the restoration difficulty and limitations of aquatic species in streams, which is the foundation of the next section.

5. Considerations for Successful Macrophyte Introduction

The authors only talk about the plant aspect, but how about the stream environment. Generally, these degradation streams are not appropriate for plant introduction.

6. Towards a successful implementation Strategy for Submerged Macrophyte Introduction

The authors did not concern the biodiversity of aquatic plants, which is not corresponding to the title.

7. Conclusion: The current status is ok.

Overall this is an interesting and useful MS for a broad reader although some tiny flaws exist. I recommend to reconsider this MS after a minor revision.

Specific points:

Lines 1415.  This sentence“This is the result of the combined 14 effect of a multitude of often anthropogenic stressors.” is too vague.

Line 45.  The last part of this sentence is unclear.  “stressors, many of which of anthropogenic nature”

Line 81.  Delete “for”.

By the way, the whole English is very fluent and excellent, I like it.

Comments on the Quality of English Language

Restore biodiversity in streams with native submerged macrophytes is a hot topic of substantial interest. Although this manuscript review some useful perspectives of biodiversity restoration in streams with native submerged macrophytes, I have some substantial concerns about the manuscript, including aspects of the presentation and some of the interpretations. Below I describe my major concerns followed by a list of more specific points.  

Major concerns:

The Abstract does not adequately indicate the general relevance of this work.  Although the first sentence indicated why this topic is important, and the last sentence should indicate the major implications for advancing ecological understanding.

The Introduction is fine at this status, but it is better if the authors can focus more about the introduction of native submerged macrophytes.

2. Threats to Freshwater Ecosystems: The largest global threats to freshwater systems can be classified into five main pressures is not good, I think Climate change should be integrated into the sixth main pressures.

3. Policies, Water Management and Restoration Efforts: Provide more Policies, Water Management and Restoration Efforts in stream restoration,especially of native submerged species. Furthermore, more regiones rather than Europe should be payed attention to.

4. Macrophyte Introduction: Macrophytes as Primary Ecosystem Engineers:

Add some information about the restoration difficulty and limitations of aquatic species in streams, which is the foundation of the next section.

5. Considerations for Successful Macrophyte Introduction

The authors only talk about the plant aspect, but how about the stream environment. Generally, these degradation streams are not appropriate for plant introduction.

6. Towards a successful implementation Strategy for Submerged Macrophyte Introduction

The authors did not concern the biodiversity of aquatic plants, which is not corresponding to the title.

7. Conclusion: The current status is ok.

Overall this is an interesting and useful MS for a broad reader although some tiny flaws exist. I recommend to reconsider this MS after a minor revision.

Specific points:

Lines 1415.  This sentence“This is the result of the combined 14 effect of a multitude of often anthropogenic stressors.” is too vague.

Line 45.  The last part of this sentence is unclear.  “stressors, many of which of anthropogenic nature”

Line 81.  Delete “for”.

By the way, the whole English is very fluent and excellent, I like it.

Reviewer 3 Report

Comments and Suggestions for Authors

I like how the article is drafted, it is short and direct in its message. The thing is, I do not know, who would be the receiver of this message. But I suppose it is not important if this can be read as a kind of "guide" for both researchers and practitioners.

Therefore, I have some comments, which may bring more wholesomeness to the paper:

In the section 2: What about the thermal pollution and does such trends in water temperature can undermine reintroduction of macrophytes or plants will help alleviate this issue? Droughts and floods may be the most important factors in freshwater systems. Maybe add a sentence more about how the submerged aquatic vegetation typically respond to such conditions, i.r., "Changes in submerged aquatic vegetation (SAV) coverage by extended drought and flood pulses" Shivers et al., 2018

In section 4: Macrophytes also delay pollutant transport, i.e., "Influence of vegetation maintenance on flow and mixing: case study comparing fully cut with high-coverage conditions" Kalinowska et al., 2023

Section 5: Is the usage of favorable conditions like occurence of boulders, tree logs etc. what was previously mentioned, can be treated as another method for successful introduction of macrophytes?

Section 6: Is there any consideration to use, instead of locally preserved species that struggle to thrive in new climate conditions, new plants from far away biotopes, but e.g., more resilient, more suitable for invertebrates or having some other benefits? Or maybe it is just worth noting that more reintroduction examples are needed to cover various factors and species combinations, maybe in insolated biotopes if there is a risk of contact with protected nature areas. 

Round 2

Reviewer 1 Report

Comments and Suggestions for Authors

Manuscript number: plants-2890529

Title: Introduction of Native Submerged Macrophytes to Restore Biodiversity in Streams

The Authors significantly improve the manuscript and clearly responded to all the comments included in the revision.

Considering that, I recommend publication of the manuscript in section: Plant Ecology, Special Issue Macrophytes in Inland Waters: From Konwledge to Management II
